# Rubella Vaccine Introduction in the South African Public Vaccination Schedule: Mathematical Modelling for Decision Making

**DOI:** 10.3390/vaccines8030383

**Published:** 2020-07-13

**Authors:** Nkengafac Villyen Motaze, Ijeoma Edoka, Charles S. Wiysonge, C. Jessica E. Metcalf, Amy K. Winter

**Affiliations:** 1Centre for Vaccines and Immunology, National Institute for Communicable Diseases/National Health Laboratory Service, Johannesburg 2131, South Africa; 2Department of Global Health, Faculty of Medicine and Health Sciences, Stellenbosch University, Cape Town 7505, South Africa; Charles.Wiysonge@mrc.ac.za; 3South African Medical Research Council Centre for Health Economics and Decision Science-PRICELESS SA, School of Public Health, Faculty of Health Sciences, University of the Witwatersrand, Johannesburg 2193, South Africa; ijeoma.edoka@wits.ac.za; 4Cochrane South Africa, South African Medical Research Council, Cape Town 7505, South Africa; 5School of Public Health and Family Medicine, University of Cape Town, Cape Town 7925, South Africa; 6Department of Ecology and Evolutionary Biology, Princeton University, Princeton, NJ 08544, USA; cmetcalf@Princeton.edu; 7Department of Epidemiology, Bloomberg School of Public Health, Johns Hopkins University, Baltimore, MD 21205, USA; akwinter@jhu.edu

**Keywords:** rubella, congenital rubella syndrome, rubella-containing vaccine, vaccine introduction strategies, age-structured rubella transmission model

## Abstract

*Background*: age structured mathematical models have been used to evaluate the impact of rubella-containing vaccine (RCV) introduction into existing measles vaccination programs in several countries. South Africa has a well-established measles vaccination program and is considering RCV introduction. This study aimed to provide a comparison of different scenarios and their relative costs within the context of congenital rubella syndrome (CRS) reduction or elimination. *Methods*: we used a previously published age-structured deterministic discrete time rubella transmission model. We obtained estimates of vaccine costs from the South African medicines price registry and the World Health Organization. We simulated RCV introduction and extracted estimates of rubella incidence, CRS incidence and effective reproductive number over 30 years. *Results*: compared to scenarios without mass campaigns, scenarios including mass campaigns resulted in more rapid elimination of rubella and congenital rubella syndrome (CRS). Routine vaccination at 12 months of age coupled with vaccination of nine-year-old children was associated with the lowest RCV cost per CRS case averted for a similar percentage CRS reduction. *Conclusion*: At 80% RCV coverage, all vaccine introduction scenarios would achieve rubella and CRS elimination in South Africa. Any RCV introduction strategy should consider a combination of routine vaccination in the primary immunization series and additional vaccination of older children.

## 1. Introduction

Rubella is a mild viral infection in children and adults but can lead to birth defects in infants born to women infected during pregnancy. These birth defects, known as congenital rubella syndrome (CRS), include transient and permanent sequelae [1]. Rubella-containing vaccines (RCV) have been in use since the late 1960s and this has led to successful elimination of rubella and CRS in many countries [2]. There is however a transient risk of increasing CRS incidence if vaccination coverage with RCVs is less than 80% because this inadequate coverage leads to reduced transmission in childhood resulting in increased number of females attaining the reproductive age group while being susceptible to rubella [3,4,5]. 

The World Health Organization (WHO) recommends introducing RCVs in countries that are aiming for measles elimination by concomitant administration of vaccines against measles and rubella [6]. Available combinations of RCV include measles-rubella (MR), measles-mumps-rubella (MMR), and measles-mumps-rubella-varicella (MMRV) vaccines. Given the goal of achieving elimination in 10 to 20 years, the WHO recommends an initial mass vaccination campaign, or supplementary immunization activities (SIAs), targeting individuals 9 months to 14 years of age, followed by introducing RCV into the routine vaccination program with regular SIAs every four to five years [6] to reach children missed by routine vaccination. This strategy is financially supported by Gavi, the Vaccine Alliance, for Gavi eligible countries. The opportunity to incorporate the RCV vaccine into existing measles immunization programs comes with substantial cost-savings relative to the more usual situation where a completely different vaccine has to be introduced, as a different target population, delivery scheme and other program mechanisms need to be developed.

A number of guiding principles have been formulated to assist countries planning RCV introduction in their public vaccination schedule [7] beyond the wide age range for SIAs coupled with routine vaccination (generally via combination with measles). Recommended steps include development of integrated surveillance for rubella and measles, follow-up SIAs, filling immunity gaps in older populations, and CRS surveillance [8].

Several countries in the WHO Africa region have introduced RCV into their Expanded Program on Immunization (EPI) schedule [9] as part of the measles-rubella elimination strategy. According to the current WHO measles and rubella strategic plan 2012-2020, measles is targeted for elimination by 2020, but the WHO African (AFRO) region has not yet set a target for rubella elimination [10]. In the current EPI schedule of South Africa, measles vaccine is given at six months and 12 months of age [11]. Introduction of RCVs is being considered and careful planning is therefore required in order to maximize benefits of the intervention.

Mathematical modelling has played an important role in the development of public health policy for rubella. This is in part because the nuanced and age-dependent nature of the burden of rubella infection requires a dynamical framework in projecting its epidemiology. In fact, delays to wide-spread introduction of RCV stemmed from mathematical models of the potential impact of vaccination introduction on the burden of CRS from the 1980s [12,13]. Further, mathematical models combined with serological surveys have been key to evaluating the burden of CRS, since the manifestation of this syndrome is hard to disentangle from many other potential aetiologies in resource poor settings [14]. Indeed, estimates of CRS incidence per 100,000 live births in Africa in the 90s obtained by applying mathematical models to age profiles of serology ranged between 104 (25 to 246) [15] and 115 (55 to 231) [14] in 1996. These estimates were similar for 2000; 116 (55 to 232) and 2010; 116 (56 to 235) [14]. 

Recently, age-structured mathematical models designed to reflect contemporary ranges of human demography have been used to re-evaluate the impact of RCV introduction into existing measles programs [16]. The basic model was further refined to more closely reflect particular settings (requiring country-specific estimates of population age distribution, birth rate, age-specific fertility rates, contact patterns, and existing vaccination strategies) to develop context-specific recommendations for RCV introduction in African and Asian countries [17,18], including South Africa [19]. This latter analysis suggested that introduction of the vaccine was likely to result in a reduction of the burden of CRS, with negligible impacts of spatial variability in vaccination coverage and transmission; but did not formally address the question of the added value of introduction of the vaccine. 

In this paper, we simulate rubella infection in South Africa using an age-structured rubella transmission model. We explore the effects of a number of vaccine introduction scenarios on patterns of rubella infection and CRS incidence, extending the range of scenarios beyond those explored by Metcalf et al. [19] to evaluate combination of RCV with Human papilloma virus vaccination, different scheduling for SIAs as well as a range of different SIA target age ranges. We provide the first comparison of costs of these scenarios (RCV cost relative to the cost of measles vaccination alone) within the context of CRS reduction or elimination over different periods of time [6]. This work can be used to guide the decision-making process when the government of South Africa introduces RCV into the public vaccination schedule.

## 2. Methods

### 2.1. Age-Structured Rubella Model

To explore the impact of introduction of RCV into South Africa, we used a previously published deterministic discrete time age-structured model [16,20] which is characterized by a matrix capturing transitions between epidemiological states (maternally immune (M), susceptible (S), infected (I), recovered (R), and vaccinated (V)) and between age groups (Appendix A). Individuals in the maternally immune (M) compartment are children born to mothers who are immune to rubella and passively acquire immunity. Susceptible (S) individuals are those who lose maternal immunity or are born susceptible and at risk of becoming infected (I). Infected individuals recover by the next time step moving into the recovered (R) compartment. The vaccinated (V) compartment represents individuals who receive RCV and are successfully immunized. The time step used in the model was ~16 days, as this corresponds to the generation time of rubella. See Appendix A for model details.

One of the key model inputs is the basic reproductive number (R_0_), which is the average number of secondary infections resulting from a typical infectious person in a totally susceptible population. The value of R_0_ used in this model was 7.9 and was obtained from a previously published modelling study estimating R_0_ for 40 African countries [21]. We proceeded to run simulations with different estimates for R_0_ in a sensitivity analysis. The highest estimate used was an R_0_ of 12 which was estimated in Ethiopia [22] and the lowest estimate estimated in Burkina Faso was 3.3 [21]. The nature of interactions between individuals influences transmission of infectious diseases. This was represented in the model as a function for seasonal amplification [23,24,25] and age-specific mixing based on estimated age-dependent social contact [26] and non-modelled heterogeneities [27]. Duration of maternal immunity [28] and vaccine efficacy [29] were estimated from published literature. Demographic data over time for South Africa were obtained from projections by the United Nations (UN) Population Division [30]. See Appendix A for model parameter details.

Preventing CRS is the main reason for administering rubella vaccination, given the mild nature of infection among children and adults. We therefore assessed the impact of vaccine introduction on both rubella and CRS incidence over time for all scenarios. To estimate the burden of CRS, we combined rubella age-specific incidence generated by the model with an age-related fertility profile for South Africa obtained from the UN Population Division 2015 estimates [30]. 

### 2.2. Vaccine Introduction Scenarios

We explored vaccine introduction scenarios that reflect options that might be implemented in South Africa (Table 1). The measles vaccine is currently administered at six months and 12 months as part of the EPI schedule in South Africa. Previously, country-wide SIAs were organized every four to five years but in recent years, SIAs are only organized as measles outbreak control measures in affected districts or provinces. On the contrary, RCVs are currently available in South Africa but only in the private health sector, which caters for about 15% of the population. We therefore fixed RCV coverage in our simulations to 15% prior to introducing the vaccine in the EPI schedule. The WHO recommends an initial SIA, targeting a wide age range of individuals, with the concurrent introduction of RCV into the routine EPI schedule [6]. We simulated rubella disease dynamics for 55 years (1995 to 2050) by first simulating endemic rubella disease dynamics (from 1995 to 2019) before initiating vaccine introduction (from 2020 onwards). Analyses covered three time horizons (10 years, 20 years, and 30 years) following RCV introduction to encompass various time frames required for CRS elimination using different RCV introduction strategies [6].

Rubella containing vaccines if introduced into the South African EPI program will be in combination with measles vaccine. Estimates of coverage for the second dose of routine measles vaccination [31] according to the South African government differ from those of WHO (79% versus 53% in 2017, 75% versus 50% in 2018). To encompass the emergent properties of a range of potential coverage values and target ages for routine immunization (9 or 12 months), we considered an array of scenarios reflecting different levels of coverage for routine vaccination achieved by 12 months, ranging from 60% to 95%. We also considered one scenario in which a dose of RCV was administered to boys and girls at the same age as the human papillomavirus (HPV) vaccine. The HPV vaccine is administered each year to nine year-old girls in schools in South Africa. It is reasonable to assume that this approach could be considered in an attempt to cover the adolescent population in the absence of SIAs. For all RCV introduction scenarios, including SIAs, we set the coverage of RCV during SIAs at 80% because this is the minimum coverage recommended by WHO [6]. We also set the coverage of RCV at 80% at the time of co-administration with HPV vaccine in order to be consistent with RCV coverage for individuals who are above the age for the primary series of RCV.

### 2.3. Evaluating Costs of RCV Introduction 

We evaluated costs relating to introducing the RCV from the perspective of the South African government as additional cost per dose of RCV compared to the current practice of administering measles-only containing vaccine. In the absence of detailed information, we assumed that no additional program costs are associated with introduction of the RCV vaccine, due to a direct substitution of the RCV with measles-only containing vaccine. Thus, for rubella, focusing on additional (undiscounted) costs relative to the measles baseline should be appropriate to guiding the investment case for rubella vaccine introduction. 

The price per dose of the measles vaccine currently used in South Africa (10 doze vial) in South African Rands (ZAR), is ZAR 29.13 [32]. For RCVs, we estimated price per dose for MR (ZAR 38.00 per dose) and MMR (ZAR 81.00 per doze) based on prices reported for the Pan American Health Organization (PAHO) in the Market Information for Access to Vaccines database [33]. PAHO prices are usually within 10% of vaccine prices in South Africa (personal communication with the national cold chain manager). We assume that a multi-year contract will be signed such that the price of the RCV remains the same for the duration of the simulations. To obtain additional costs of RCV introduction, the difference in price per dose between the RCVs (MR and MMR) and the measles vaccine was multiplied by the total number of persons vaccinated under each scenario. Total numbers of persons vaccinated under each scenario were estimated by applying expected coverage estimates to corresponding target populations obtained from the UN population estimates [30]. The number of CRS cases averted in each scenario was obtained by subtracting the number of CRS cases in that scenario from the number of CRS cases in scenario 1.

For each scenario, we calculated the number of RCV doses per CRS case averted by dividing the total number of RCV doses used by the total number of CRS cases averted. The corresponding cost per CRS case averted was obtained by dividing additional RCV costs by CRS cases averted. These estimates were obtained for MR and MMR using 60% coverage representing the worst case scenario, 80% coverage representing the WHO recommended minimum coverage for RCV introduction and the 95% coverage level representing the best case scenario.

### 2.4. Evaluating DALYs Averted by RCV Introduction

To assess total undiscounted disability-adjusted life years (DALYs) averted by the introduction of RCV, we estimated the number of CRS cases averted from 2020 to 2050 (as the difference in CRS cases between each RCV scenario and the no RCV scenario) and applied this to undiscounted DALYs lost per CRS case. DALYs lost per CRS case were obtained from an existing study reporting DALYs lost for a range of countries using disability weights from the 1990 and 2010 Global Burden of Disease (GBD) study [34]. For the purposes of this paper, we use estimates of DALYs lost reported for upper middle-income countries (World Bank classification for South Africa) using 2010 GBD disability weights.

### 2.5. Evaluating Impact on Outbreak Risk

An important measure of the success of vaccination programs is the degree to which they can sustain elimination and (even transiently) prevent outbreaks [35]. The effective reproductive number (R_E_) is the average number of secondary cases resulting from the introduction of one infectious person into a population containing some individuals who are not susceptible to the infection [36]. Estimates of R_E_ have been used to inform timing of vaccination interventions for preventing disease outbreaks [37], and to determine the likelihood for disease outbreaks in populations if an infectious case was introduced [38].The endemic nature of rubella in the absence of RCV and the subsequent change in number of susceptible individuals with an introduction of RCV could result in a change in R_E_ over time. A value for R_E_ greater than one implies rubella outbreaks can occur and values less than 1 mean that the infection goes into extinction. Values of R_E_ over time were extracted from the simulations to understand the impact of various scenarios on estimated time to rubella elimination and periods when there was a rebound in R_E_ from values below 1 to values greater than 1. These fluctuations in R_E_ associated with different scenarios will inform vaccination activities which should be implemented even after perceived short-term elimination is achieved to avoid possible rebound in rubella incidence. We extracted and presented values of R_E_ for the entire period during which the simulations were run. The effective reproduction number (R_E_) was estimated from the model output using the next generation method [39]. 

## 3. Results

### 3.1. Rubella Incidence

Figure 1 represents the typical patterns we see across the 6 vaccination campaigns. In the absence of rubella vaccination in the public sector (scenario 1), rubella remains endemic with annual peaks in incidence (Figure 1A). There is a decrease in the incidence of rubella over time due to declining birth rates which decreases the rate at which individuals become infected in the population resulting in an increase in the average age of infection (Figure 1). For all scenarios with a mass campaign (scenarios 3, 4, and 5), there is a sharp decrease in rubella incidence (Figure 1C–E). For these same scenarios, we see an increase in the average age of infection; however, this is only among very few to no rubella cases, so is not meaningful when evaluating the impact of vaccination. For scenarios with RCV introduction without mass vaccination (scenarios 2 and 6), there is a gradual decrease in rubella incidence as well as a gradual increase in average age of infection (Figure 1B,F). The higher average age of infection with decreased rubella incidence results from higher relative numbers of rubella in individuals of older age groups compared to cases in children that substantially reduce following vaccine introduction.

### 3.2. CRS Incidence

Without RCV introduction, CRS incidence (CRS cases per 100,000 live births) increased steadily over time as a result of rising average age of infection (black line, Figure 2A). Introduction of RCVs in the EPI schedule along with an initial mass campaign leads to rapid reduction in the incidence of CRS while this reduction was much slower when RCVs were introduced without an initial mass campaign (Figure 2C–E versus Figure 2B,F). In scenario 2, CRS incidence initially drops following introduction of RCV and subsequently begins to increase; the timing of increase vary with RCV coverage levels. Lower levels of RCV coverage lead to a shorter time to a recrudescence of CRS cases following an observed initial decrease (Figure 2B). In scenarios 3 and 4, low RCV coverage (60%) leads to a decrease in the incidence of CRS cases followed by a slight recrudescence of cases after 20 years (Figure 2C,D). This was not observed for scenario 5 even with RCV coverage levels as low as 60% due to the frequent campaigns occurring every five years and obtaining 80% coverage (Figure 2E). For scenario 6, there was also no recrudescence of CRS incidence regardless of RCV coverage but the decrease in CRS incidence was faster than scenario 2 and slower compared to scenarios 3, 4 and 5 (Figure 2F). A reduction in CRS incidence to less than 1 per 100,000 live births was achieved and sustained for RCV coverage values of 80% and above for all scenarios. The time to CRS elimination was shortest and did not vary with R_0_ for scenarios 3 through 5 at 80% RCV coverage (Appendix A). For scenario 6, CRS elimination following RCV introduction was quicker with a higher values of R_0_ and time to CRS elimination was shorter in scenario 6 compared to scenario 2 (Appendix A). 

### 3.3. CRS Cases Averted

The number for CRS cases averted in scenarios 2–6 compared to scenario 1 over different time horizons (10 years, 20 years, and 30 years) is shown in Figure 3. The number of CRS cases averted was consistently smallest for scenario 2 regardless of vaccine coverage. The highest incremental number of cases averted was observed between 60% coverage and 80% coverage. There is little difference in CRS cases averted between scenarios 3, 4, and 5 for 80% RCV coverage compared to scenarios where RCV coverage was 95%. For scenario 6, there were fewer CRS cases averted compared to scenarios 3, 4, and 5 but more cases averted compared to scenario 2. 

### 3.4. Efficiency of RCV to Reduce CRS Cases

We present comparisons of the percentage reduction in CRS cases with the number of RCV doses per CRS case averted and the corresponding additional vaccine cost per CRS case averted (Figure 4). The cost of the MR (Figure 4A) vaccine is lower than of the MMR vaccine (Figure 4B), as such the cost per CRS case averted is also lower. As observed above, the highest percent reduction in CRS cases is observed in scenarios 3, 4, and 5 within all coverage levels and across time horizons (Figure 3). At higher levels of coverage (80–95%), scenarios 3, 4, and 5 have the highest number of doses per case averted and consequently, the highest RCV cost per case averted. However, at the lowest coverage level (60%), scenario 6 is observed to have the lowest number of doses per CRS case averted and the lowest RCV cost per case averted across all study time horizons. Within each study scenario, number of doses and cost per CRS averted increases with coverage. An exception is observed in scenario 2 where at 60% coverage, a higher number of doses and cost per case averted is observed compared to 80% and 95%. This can be explained by re-emerging CRS cases observed at lower levels of coverage in scenario 2 approximately 5 years post-RCV introduction (Figure 2A). When RCV cost per case averted is compared to the additional benefits of each scenario (represented here as percent CRS reduction), scenario 6 (at 60% coverage) is observed to have the least RCV cost at a high percent reduction in CRS cases across all time horizons. 

The relationship between RCV and CRS cases averted across the scenarios and vaccination coverages described above is qualitatively similar to the relationship between RCV and undiscounted DALYS averted (Appendix A). The maximum DALYs averted in any scenario is 285,611 based on 12,472 CRS cases averted (Appendix A). 

### 3.5. Effective Reproductive Number (R_E_)

We present values of R_E_ for all scenarios, maintaining RCV coverage at 80% (Figure 5). For scenario 1 (absence of RCV introduction) the value of R_E_ fluctuates around 1.2 which consistent with the periodic peaks in rubella cases described in South Africa [19]. In scenario 6, R_E_ drops to values below one over about 5 years and stays below one over the entire simulation period. In scenario 5, R_E_ dropped immediately and remained below one over the entire simulation period. In scenarios 3, 4, and 5, the drop in R_E_ was followed by a brief rebound before a subsequent decrease as a result of accumulating susceptible individuals. The drop in R_E_ to below 1 was much slower in scenario 2 (over 13 years) and this represents a prolonged period during which outbreaks could occur. The result that R_E_ drops and stays below one is robust to routine coverage level for all scenarios so long as RCV coverage is 65% or higher (Appendix A). At 60% coverage, the R_E_ drops below one and then increases slowly eventually crossing above one in scenarios 2, 3, and 4 (Appendix A). At 80% RCV coverage, R_E_ drops and remains at values below one for all scenarios, although this drop is immediate for scenarios 3 through 5 and gradual for scenarios 2 and 6. The time required for R_E_ to drop below one was inversely related to R0 values and was longer for scenario 2 compared to scenario 6 (Appendix A). 

## 4. Discussion

The incidence of CRS is used as a measure for evaluating the effectiveness of rubella vaccination strategies and CRS elimination is a major milestone of RCV introduction. Different vaccine introduction strategies achieve CRS elimination over various periods of time depending on the vaccination coverage achieved and target age groups for vaccination [6]. In the absence of RCV introduction into the EPI schedule, CRS incidence increased steadily over the simulation period. A previous modelling study used results of rubella antibody testing and estimated that CRS incidence in South Africa for 2005 ranged from 16 to 69 CRS cases per 100,000 live births. When exploring the effect of varying RCV coverage levels (60% through 95%) on CRS incidence, our study shows that a reduction in CRS incidence to less than 1 per 100,000 live births can be achieved and sustained when RCV coverage was at least 80% for all scenarios.

Our results show that introducing RCV into the EPI schedule without mass campaigns or SIAs (scenarios 2 and 6) leads to a slower decrease in rubella cases and CRS incidence over time compared to an abrupt decrease observed in scenarios with SIAs. Introducing the vaccine into the routine EPI program protects infants and reduces rubella virus circulation but leaves unprotected individuals of reproductive age who remain susceptible to rubella infection and subsequently CRS. On the other hand, RCV introduction accompanied by a mass campaign targeting individuals up to 15 years of age (scenarios 3–5) results in an immediate reduction of rubella and CRS incidence. This is due to greater reduction in rubella virus circulation in the population and although individuals above 15 years of age are not vaccinated, the reduction in viral circulation and subsequently rubella cases is sufficient to lead to elimination of CRS.

The average age of infection for rubella shifts from the younger to the older age groups following RCV introduction as rubella incidence declines. Prior to RCV introduction, the bulk of rubella infections occur in childhood consistent with a lower average age of infection. A decrease in the number of infections in the younger age group targeted by RCVs results in older individuals bearing a higher relative burden of rubella infection. 

Overall, cumulative CRS cases averted was highest in scenarios with higher number of mass campaigns (scenarios 5, followed by scenario 4 and finally scenario 3). The number of CRS cases averted in scenario 6 was lower than in scenarios 3 through 5, but higher than in scenario 2. This suggests that in the absence of mass campaigns, targeting children of specific age groups might be a reasonable alternative. When vaccination coverage are low, we show that reduction in rubella virus circulation can be achieved by vaccinating individuals up to 15 years of age, accompanied with multiple SIA every 5 years (scenario 5). This is sufficient to lead to CRS elimination at coverage levels as low as 60%. However, it is advisable for governments to aim for at least the 80% threshold recommended by WHO, or better still, 95% coverage which is required for measles elimination since the combined vaccine will be used. Achieving high levels of vaccine coverage is important when considering other RCV introduction scenarios due to a possible re-emergence of CRS cases. In 2, 3, and 4, CRS incidence rises above 1 per 100,000 births a few years after RCV introduction when coverage levels are lower than 80%. For scenario 5 in which several SIAs occur after the initial RCV introduction, CRS elimination is sustained even with the lowest vaccine coverage simulated (60%). Interestingly, in all RCV introduction scenarios, the incidence of CRS remain below pre-vaccine incidence estimates irrespective of RCV coverage. This implies that all of the RCV introduction scenarios evaluated in this analysis result in lower annual CRS incidence rate compared to the current situation in South Africa of not having a RCV within the national EPI program. Nonetheless, the WHO recommends that RCV should be introduced when countries meet the 80% threshold and highlights the importance of mass campaigns targeting older children in addition to routine vaccination of infants, not only at introduction of RCV but also as regular follow-up mass campaigns.

Mass campaigns have high budget implications. We found that although scenarios 4 and 5 have the highest percent reduction in CRS cases, the costs associated with both scenarios are higher compared to other scenarios. Conversely, the lowest cost per case averted is observed in scenario 6 (vaccinating only infants and nine year olds with no SIAs or mass campaigns) when coverage levels are low at 60%. However, this does not account for additional costs associated with managing re-emergent CRS cases, 5–10 years post introduction of RCV. Therefore, the additional benefits (CRS cases averted) observed in each scenario need to be weighed against additional RCV costs and CRS management costs in an economic evaluation to inform the most cost-effective scenario that can be implemented in South Africa.

In South Africa, the human papillomavirus vaccine is administered to school-going girls at nine years of age [11] and this is an attractive option for reducing rubella susceptibility (and hence reduction in CRS incidence) in adolescent females. Implementing a RCV dose for nine-year-old children would require scaling up of the school vaccination program to account for both girls and boys, a move that would require additional resources since this would be an annual intervention. We simulated this option and it resulted in lower RCV cost for a comparable percentage CRS reduction compared to all other scenarios. It is important to note however, we assumed that all children age nine years old are enrolled and therefore eligible for the vaccine. Enrolment less than 100% will result in a lower impact of RCV on CRS incidence in this scenario. The additional RCV cost alone (compared to the cost of the current measles vaccine) might not be adequate to assess the cost-effectiveness of this strategy; however, it does provide a preliminary insights into the comparative costs of the vaccination scenarios modelled. 

The cost of RCV increased with increasing target age group for SIAs. This was inversely proportional to the CRS burden since the higher number of vaccinated individuals inversely correlates with the number of susceptible individuals with resulting decrease in CRS incidence. Routine vaccination coupled with vaccination of nine years old children (scenario 6) achieved reductions in CRS cases at the least vaccine costs per CRS case averted compared to the other scenarios modelled. Scaling up HPV vaccination in schools to include boys for the RCV component appears to be a sustainable option. South Africa is a middle-income country and is not eligible to receive Gavi funding so the costs of RCV introduction will be entirely borne by the national government. The estimates of vaccine cost from the government perspective indicate higher costs with increasing target age for mass campaigns but the increasing cost corresponds to decreasing CRS incidence. A trade-off will have to be made by decision makers regarding this. For RCV, most of the added cost (when compared to the current measles vaccine) is tied to the additional cost of the rubella component of the vaccine since routine measles vaccination is an established program and mass campaigns can be organized, but with additional costs. Although vaccine wastage and other program-related costs were not estimated, it is likely that vaccine wastage from a 10-dose RCV formulation will not change from current levels of the measles vaccine. Further considerations will have to be made if any formulation other than the 10 dose vial is used.

Keeping RCV coverage for routine immunization and SIAs at 80%, the effective reproduction number (R_E_) dropped sharply from values fluctuating around 1.2 in the pre-vaccine era to less than 1 following vaccine introduction except for scenarios 2 and 6 in which R_E_ dropped to values below one over several years. This delay could lead to surges in rubella cases and eventually CRS cases. With the exception of scenario 5, scenarios with an initial SIA are associated with a rise in values of R_E_ after the initial drop. The rebound increase in R_E_ reflects growth in the susceptible population as a result of accumulation of successive fractions of the birth cohort that are unvaccinated each year. As a result, subsequent mass campaigns cover some of these missed individuals, as is the case in scenarios 4 and 5, causing the value of R_E_ to drop followed by a progressive rise. High RCV coverage levels should be maintained to keep R_E_ below 1, thereby avoiding rubella outbreaks that could lead to CRS cases.

## 5. Strengths and Limitations

A major strength of this paper is the fact that the age-structured model has been used to inform national RCV introduction strategies in several countries. Secondly, the choice of scenarios simulated and model inputs were informed by sources relevant to the local setting which enables better estimations. Lastly, the approach used to estimate additional vaccine introduction costs of RCV compared to monovalent measles vaccine can be applied to other countries with a similar immunization schedule for measles and rubella. The main limitation of this study is the fact that the model produces outputs for the entire South African population and does not account for disparities between geographic units such as provinces or districts. Factors such as contact patterns, birth rates, and vaccine coverage could differ between districts or provinces, leading to variation in local disease dynamics. Additionally, the vaccine scenarios assume individuals in the target age group are accessible and can be vaccinated per the assumed coverage and taking into account vaccine efficacy. Finally, all costs associated with RCV scenarios modelled in this study where not fully accounted for, limiting the use of our results in informing the prioritization of RCV scenarios on the basis of their cost-effectiveness. 

## 6. Conclusions

The output from the age-structured model emphasizes on the importance of maintaining a high vaccination coverage when introducing RCVs in the South African EPI schedule. The threshold coverage of 80% should be maintained for all vaccine introduction scenarios to achieve rubella and CRS elimination while attaining 95% could, in addition, lead to measles elimination. The results also support a vaccine introduction strategy that entails a combination of routine RCV vaccination in the primary immunization series and additional vaccination of older children in order to maximize the impact on rubella and CRS. More robust economic evaluation studies would be required to inform the prioritization of RCV introduction strategies in South Africa.

## Figures and Tables

**Figure 1 vaccines-08-00383-f001:**
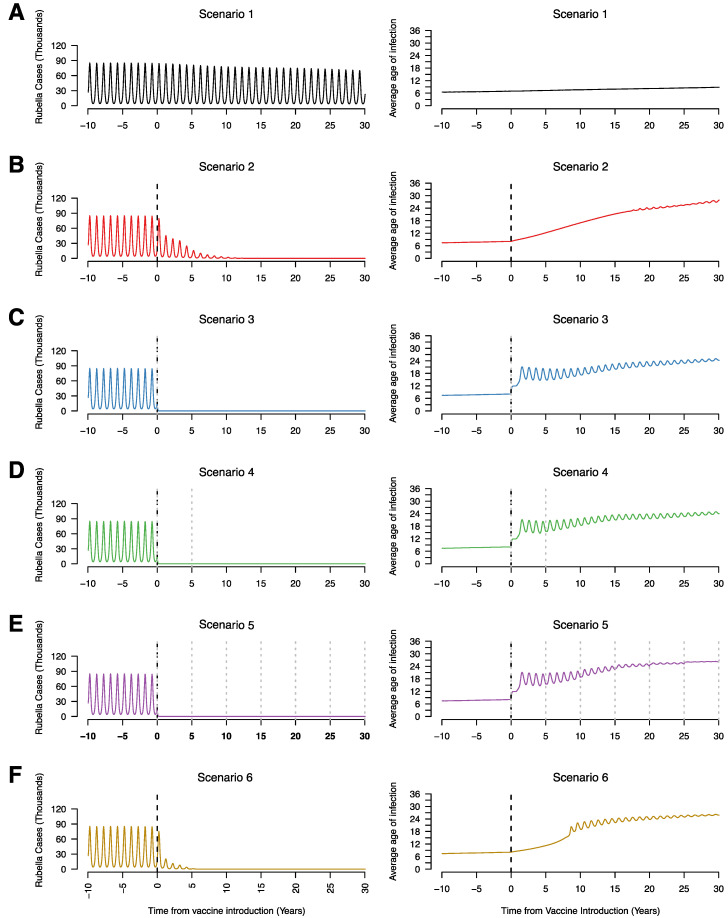
Time series of the annual number of rubella infections (in thousands) and average age of infection for scenario 1 (**A**), scenario 2 (**B**), scenario 3 (**C**), scenario 4 (**D**), scenario 5 (**E**), and scenario 6 (**F**), assuming 80% RCV coverage in routine and campaigns if relevant to the scenario. The vertical black dotted line represents the year of RCV introduction (year zero) and the grey dotted lines represent supplementary immunization activities (SIAs).

**Figure 2 vaccines-08-00383-f002:**
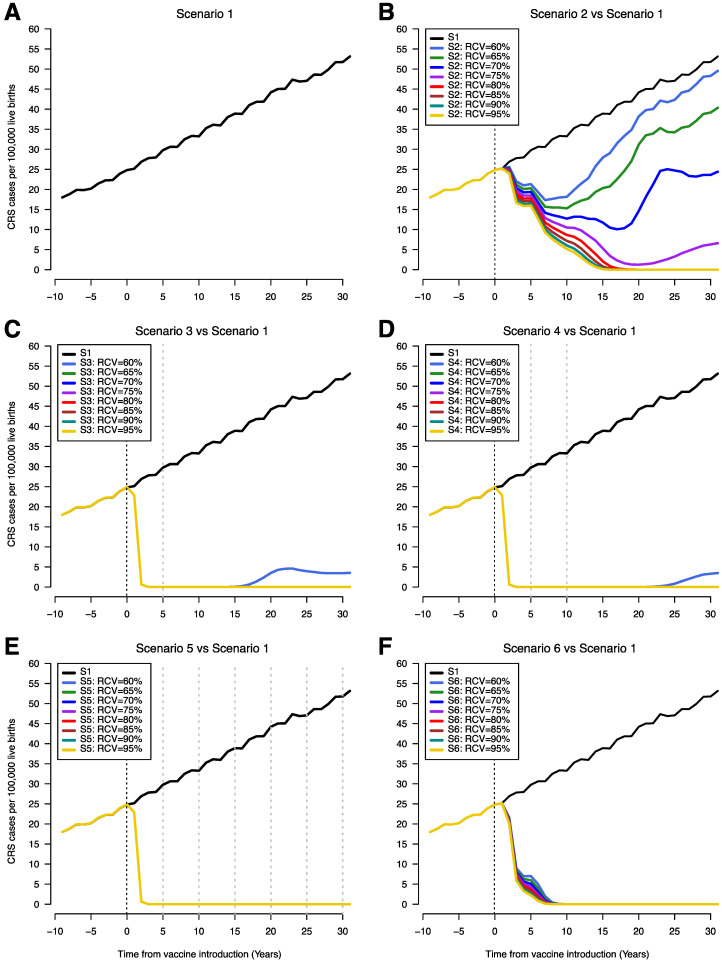
Time series of congenital rubella syndrome (CRS) incidence (CRS cases per 100,000 live births) showing scenario 1 (**A**) and comparing scenario 1 with scenarios 2–6 (**B**–**F**). The vertical black dotted line indicates year of RCV introduction (year zero) and the grey dotted lines represent SIAs. The x-axis shows time from 10 years prior to RCV introduction to 30 years after RCV introduction. CRS incidence estimates overlap on the plots for different RCV coverages, so that only the line for 95% coverage appears on the graph (RCV coverages 65–95% years 0–30 (**C** and **D**), RCV coverages 60-95% years 0–30 (**E**), and RCV coverages 60–95% years 8–30 (**F**))**.** (S1 = scenario 1, S2 = scenario 2, S3 = scenario 3, S4 = scenario 4, S5 = scenario 5, S6 = scenario 6).

**Figure 3 vaccines-08-00383-f003:**
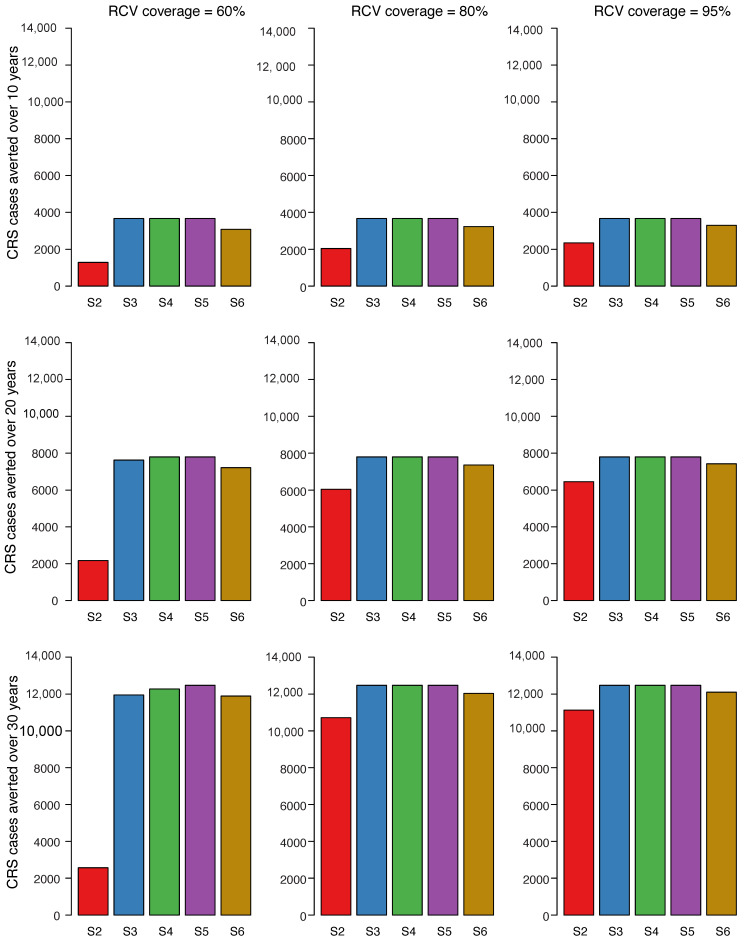
Cumulative number of CRS cases averted for scenarios 2 through 6 compared to scenario 1 over three time horizons post vaccine introduction. Each row of figures represents a different time horizon; 10 years (**A**), 20 years (**B**), and 30 years (**C**). Each column represents a different level of vaccination coverage for routine doses (60%, 80%, and 95%) from left to right, while maintaining coverage at 80% for vaccines administered outside the routine schedule. Scenarios 2 through 6 were each compared to scenario 1 and represent the bars in each plot (S2 = scenario 2 vs. scenario 1, S3 = scenario 3 vs. scenario 1, S4 = scenario 4 vs. scenario 1, S5 = scenario 5 vs. scenario 1, S6 = scenario 6 vs. scenario 1).

**Figure 4 vaccines-08-00383-f004:**
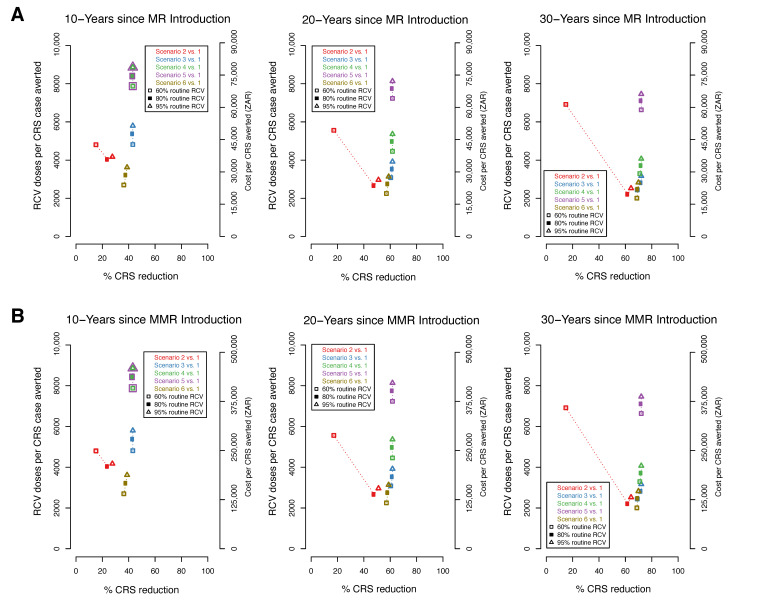
The percent reduction in CRS cases by the number of RCV doses (left y-axis) and cost (right y-axis) per CRS cases averted over three time horizons (10, 20, 30 year). Each figure compares the percent reduction in CRS cases for scenarios 2–6 to scenario 1 (represented by colours) for three RCV coverages (60%, 80%, 95% represented by shapes). Cost is evaluated based on the measles-rubella (MR) (**A**) and measles-mumps-rubella (MMR) (**B**) vaccine. At the 10 year horizon in both 4A and 4B, the numbers corresponding the scenario 4 and 5 overlap and have been represented as overlapping points.

**Figure 5 vaccines-08-00383-f005:**
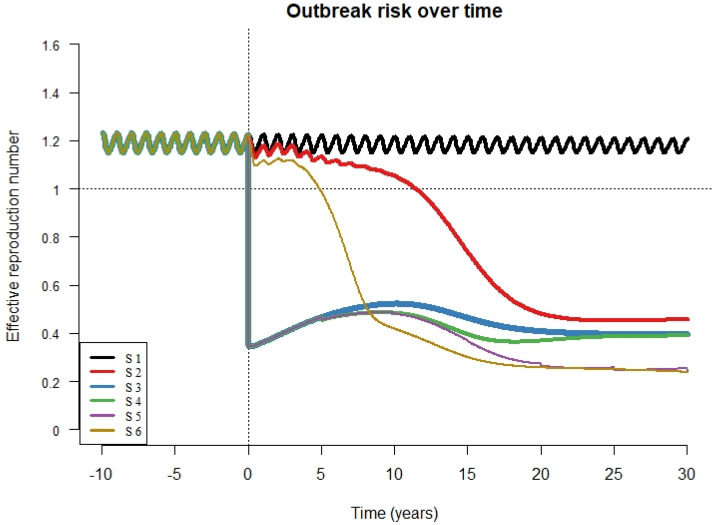
R effective (R_E_) over time for all scenarios (S1 = scenario 1, S2 = scenario 2, S3 = scenario 3, S4 = scenario 4, S5 = scenario 5, S6 = scenario 6) with vaccine coverage for both routine immunization and campaigns (if relevant to the scenario) set at 80%. The X axis shows time from 10 years prior to RCV introduction to 30 years after RCV introduction. The vertical dotted line represents the year of RCV introduction and the horizontal dotted line represents the value for the effective reproductive number above which outbreaks are likely to occur.

**Table 1 vaccines-08-00383-t001:** Possible scenarios for rubella-containing vaccine (RCV) introduction in South Africa.

Scenario	Routine Vaccination in Expanded Program on Immunization (EPI)	Target Age Group for Routine Vaccination	Target Age Group for Initial Mass Campaign	Follow-Up Mass Campaigns	
				Target Age Group	Timing
1	No RCV in EPI				
2	RCV introduction	1 year	No initial campaign	No follow-up campaign	N/A
3	RCV introduction	1 year	1 to 14 years	No follow-up campaign	N/A
4	RCV introduction	1 year	1 to 14 years	1 to 4 years	One follow-up campaign 5 years after initial campaign
5	RCV introduction	1 year	1 to 14 years	1 to 4 years	Six follow-up campaigns every 5 years after initial campaign for 30 years
6	RCV introduction	1 year and 9 years	No initial campaign	No follow-up campaign	N/A

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
