# Peer review of "Rubella Vaccine Introduction in the South African Public Vaccination Schedule: Mathematical Modelling for Decision Making"

_vaccines, 2020, doi:10.3390/vaccines8030383_

Round 1

Reviewer 1 Report

This MS dealt with different scenarios obtained from deterministic discrete time age-structured model and their relative costs within the context of rubella infection and CRS case reduction.  It is of interesting and worth for publication in this journal.

There are very few comments arisen for minor revisions.

In figures, individual graphs need be sub-numbered for better understanding.

In particular, description in results of figure 2 was difficult to understand.

It need be avoided redundancy since there are many complex factors regarding vaccine strategy.  For example, lines 413-416: this sentence is considered not to be comprehendible conjecture.

Author Response

Thank you very much for reviewing the article and providing valuable feedback.

Below are my responses to the points raised:

  1. In figures, individual graphs need be sub-numbered for better understanding.

All graphs in each figure have been numbered as requested

  1. In particular, description in results of figure 2 was difficult to understand.

We have modified the description of figure 2 to make it easier to understand from lines 253-267

  1. It need be avoided redundancy since there are many complex factors regarding vaccine strategy. For example, lines 413-416: this sentence is considered not to be comprehendible conjecture

We have modified lines 413-416 and several other sentences throughout the manuscript to improve on clarity.

Reviewer 2 Report

Please, see PDF file attached.

Author Response

Thank you very much for reviewing the article and providing valuable feedback.

Below are my responses to the points raised:

  1. This reviewer is mathematical modeler and missed details about the model. It seems confusing the model used. In different parts of the paper the authors refer to [16], [19] and [20] as sources of the model. The authors would be welcome to write a paragraph with details about the model. In [20] is where the model seems to be better described, however is only a theoretical description. How the model has been applied to the studied case? With which data? Which are the age groups used? Which are the values of the model parameter values? Which are the calibrated model parameters? How does the calibration was performed? The vaccine effectiveness has been supposed to be 100%? Why do the authors use age groups if they always show aggregated results in the graphs?
  2.  

Thank you for your request. We have provided further details of the model by making some small clarifying revisions in the main text (lines 115-138). We have additionally added a full description of the model and model parameters in an expanded Supplement 1. We believe we better answer all of your questions in the expanded Supplement 1. In terms of your last question however, we use an age-structured model because non-linear transmission dynamics are impacted by age-specific parameters (e.g., vaccine effectiveness, loss of maternal immunity, contact rates, and death rates). Additionally, our evaluation of vaccination impact is dependent on age-specific model output (e.g., CRS risk is estimated from age specific risk of infection).

  1. Which R0, and consequently, transmission rate, has been used? The authors talk about R0 in page 3, lines 124−133, taking values of 3.3, 7.9 and 12. The difference between 3.3 to 12 implies that a coverage of 70% could be enough or it is necessary more than 90%.

The main simulations were run with an R0 of 7.9 which is the value estimated for South Africa by a previous study. We explored extreme values of R0 but only simulated vaccine coverage of 80% for these sensitivity analyses.

  1. In page 6 (line 225), the authors say ”... accompanied by an increase in the average age of infection (Figure 1).” Why? Women get pregnant older?

Declines in birth rate in a simple model is functionally equivalent to decreasing the rate of transmission of infection. As the rate of transmission decreases, the rate at which susceptibles become infected (i.e. the force of infection) also decreases, resulting in an increase in the overall age distribution of infected individuals. We have revised this sentence to add this explanation.

  1. In page 6 (line 227), the authors say ”... however, this is only among very few to no rubella cases, so is not meaningful.” Maybe, however the oscillations returned by the model are very regular. To say that they are meaningful, it would be expected a model output more irregular. The authors should comment that.

You make a good point. However, in this particular sentence we are interested in the impact of vaccination on aggregated rubella cases, rather than oscillating dynamics. We have revised this sentence to specify how it is not meaningful in this particular interpretation.

  1. In Figure 5, page 12, the drop appearing in scenarios 3, 4 and 5 should be because of the vaccination. But, why the rebound?

The effective reproductive number depends on the proportion of susceptible individuals. Following the first mass campaign targeting individuals up to 14 years, there is a sharp drop in the proportion of susceptible individuals and the effective reproductive number. As new cohorts of susceptible children are born, there is an accumulation of susceptible individuals and increase in effective reproductive number until the next follow-up mass campaigns when there is another drop in susceptible individuals. However, the drop after the follow-up mass campaigns is more progressive since only children up to 4 years old are targeted and the relative drop in proportion of susceptible individuals is smaller. We added a brief explanation of this rebound in the paragraph discussing Figure 5.

  1. In Figure S2, page 18, the authors should explain why the CRS incidence is greater for R0 = 3.3 than for R0 = 12 (black lines) when the greater R0 the higher transmission rate.

The incidence of CRS is higher for lower values of R0 because the rate of infection is lower. As a result, individuals are becoming exposed to the pathogen later in life and therefore there is a higher average age of infection. A higher average age of infection means more women are ageing into their reproductive years still susceptible and at risk of infection and therefore this is a higher incidence of CRS. We revised the text in Supplement 2 to clarify this relationship.

Finally, this reviewer would like to suggest the authors for, maybe, future works, to realize that the seasonality of the disease is not necessarily due to a seasonal behavior of the transmission rate. Somme publications are appearing in this sense, as doi:10.1016/j.physa.2015.12.153, doi:10.1016/S01672789(00)00187-1.

Thank you very much for sharing this interesting reference.